# Assessment of mouse VEGF neutralization by ranibizumab and aflibercept

**Yusuke Ichiyama** *, **Riko Matsumoto**, **Shumpei Obata**, **Osamu Sawada,**
**Yoshitsugu Saishin, Masashi Kakinoki, Tomoko Sawada, Masahito Ohji**

Department of Ophthalmology, Shiga University of Medical Science, Seta Tsukinowacho, Otsu, Shiga, Japan

* ichiyama@belle.shiga-med.ac.jp

## Abstract

### Purpose

To assess the interaction between ranibizumab, aflibercept, and mouse vascular endothelial growth factor (VEGF), both in vivo and in vitro.

### Methods

In vivo, the effect of intravitreal injection of ranibizumab and aflibercept on oxygen induced retinopathy (OIR) and the effect of multiple intraperitoneal injections of ranibizumab and aflibercept on neonatal mice were assessed. In vitro, the interaction of mouse VEGF-A with aflibercept or ranibizumab as the primary antibody was analyzed by Western blot.

### Results

In both experiments using intravitreal injections in OIR mice and multiple intraperitoneal injections in neonatal mice, anti-VEGF effects were observed with aflibercept, but not with ranibizumab. Western blot analysis showed immunoreactive bands for mouse VEGF-A in the aflibercept-probed blot, but not in the ranibizumab-probed blot.

### Conclusions

Aflibercept but not ranibizumab interacts with mouse VEGF, both in vivo and in vitro. When conducting experiments using anti-VEGF drugs in mice, aflibercept is suitable, but ranibizumab is not.

## Introduction

Intravitreal anti-VEGF injection has been a first line choice for the treatment of major retinal vascular diseases including age-related macular degeneration [1–3], diabetic macular edema [4, 5], macular edema due to retinal vein occlusion [6–8], myopic choroidal neovascularization [9], and retinopathy of prematurity [10, 11]. Currently, bevacizumab, ranibizumab, aflibercept, brolcizumab and faricimab are the major anti-VEGF drugs in ophthalmic practice

**Funding:** This work was supported by grants-in-aid for Scientific Research on Innovative Areas from the Ministry of Education, Culture, Sports, Science, and Technology of Japan (19K18877). This funding organization had no role in the design or conduct of this study.

**Competing interests:** The authors have declared that no competing interests exist.

[12]. Accordingly, bevacizumab [13–16], ranibizumab [17–23], and aflibercept [24–28] are frequently used in experiments with mouse models of retinal vascular diseases. Notably, Ferrara N et al. reported that bevacizumab's ability to neutralize VEGF was species specific and lacking in mice [29, 30]. It indicates that some types of anti-human VEGF antibody may be unsuitable for experiments in mice. Ranibizumab, a humanized monoclonal antibody Fab fragment, has the same structure as bevacizumab Fab except for five amino acids although ranibizumab and bevacizumab were generated from different anti-VEGF Fab fragments [31]. Aflibercept is a recombinant fusion protein, and its VEGF-binding portion consists of the extracellular domains from "human" VEGF receptors 1 and 2 [32]. Therefore, it is possible that ranibizumab and aflibercept may lack the ability to neutralize mouse VEGF as bevacizumab. To the best of our knowledge, however, there is no report focusing on the interaction between ranibizumab, aflibercept, and mouse VEGF. The purpose of this study is to assess the interactions of mouse VEGF with ranibizumab or aflibercept, both in vivo and in vitro.

## Materials and methods

### Mice and tissue collection

All animal experiments were performed using C57BL/6J mice, their use for experimental purposes was approved by the Research Center for Life Sciences, Shiga University of Medical Science in accordance with institutional and governmental guidelines (Permit Number: 2021-4-8), and all animal procedures were adherent to the ARVO Statement for the Use of Animals in Ophthalmic and Vision Research. Euthanasia for tissue sample collection was achieved by performing cardiac puncture exsanguination under isoflurane anesthesia. The right atrium was then pierced and 5 mL of phosphate-buffered saline solution (PBS) was perfused through the left ventricle. Right after, 5 mL of 4% paraformaldehyde (PFA) at 4°C was perfused in the same manner to start fixation of the tissues [28]. A total of 46 mice were used for this study.

### Oxygen induced retinopathy model and intraocular injection

An oxygen induced retinopathy (OIR) model using C57BL/6J mouse neonates was produced as previously described [28]. Briefly, at postnatal day (P) 8, mice with nursing mothers were exposed to 85% oxygen for 3 consecutive days. At P11, the mice were removed from the oxygen chamber and returned to room air. Intraocular injections of 0.5 μL of PBS, 20 μg of aflibercept (0.5 μL of Eylea®, Bayer AG, Leverkusen, Germany) or 5 μg of ranibizumab (0.5 μL of Lucentis®, Novartis Pharma K.K., Tokyo, Japan) into the vitreous body were performed under isoflurane anesthesia at P12 using 33-gauge needles (ITO Corporation, Shizuoka, Japan). To minimize the effects of individual differences, littermates were divided into the above three groups. In our previous study [28], we analyzed OIR retinas 2 days (P14) and 4 days (P16) after intravitreal aflibercept injections and found that the anti-VEGF effects of aflibercept were stronger on P14 than on P16. Therefore, we analyzed OIR retinas 2 days after injections (P14) in this study.

### Multiple intraperitoneal injections in neonatal mice

PBS, 10 mg/kg of aflibercept, or 2.5 mg/kg or 10 mg/kg of ranibizumab was intraperitoneally injected into C57BL/6J mouse neonates daily from P3 to P6. To minimize the effects of individual differences, littermates were divided into the above four groups. The mice were weighed at all time points of injection. At P6, eyes and kidneys were harvested 6 hours after the last injection.

## Immunohistochemistry

Whole-mounted retinas and pupillary membranes were produced according to previous reports [28, 33]. Enucleated eyes were fixed for 30 minutes in 4% PFA at room temperature. A small hole was made in the cornea using a 27-gauge needle, and a circular incision was made using fine scissors. Then, retinal cups and pupillary membranes were dissected from the eyes and postfixed for 30 minutes in 4% PFA at 4˚C. Kidneys were fixed overnight in 4% PFA, snap-frozen in optimal cutting temperature (OCT) compound (Sakura Finetek USA, Inc., Torrance, CA, USA), and sectioned to a thickness of 12 μm.

The primary antibodies were biotinylated isolectin B4 (iB4, B-1205; 1:500; Vector Laboratories Inc., Burlingame, CA, USA), anti-CD31 (ab119341; 1:1000; Abcam plc, Cambridge, UK), anti-Ter119 (MAB1125; 1:250; R&D Systems, Inc.), anti-collagen IV (2150–1470; 1:500; Bio-Rad Laboratories, Inc.), and anti-Erg (ab92513; 1:2000; Abcam plc, Cambridge, UK). The secondary antibodies were suitable species-specific secondary antibodies (Jackson ImmunoResearch Laboratories, Inc., West Grove, PA, USA) or streptavidin coupled to Alexa Fluor dyes (Invitrogen, Waltham, MA, USA). For nuclear staining, specimens were treated with DAPI (Santa Cruz Biotechnology, Inc., Dallas, TX, USA).

## Perfusion with FITC-dextran

FITC-dextran perfusion was performed as previously described [20]. Briefly, 1 mL of PBS containing 50 mg of fluorescein-dextran (Sigma-Aldrich, St. Louis, MO) was injected into the left ventricle under deep anesthesia. Eyes were enucleated and fixed for 40 minutes in 4% PFA at room temperature. A small hole was made in the cornea using a 27-gauge needle, and a circular incision was made using fine scissors. Retinal cups were then dissected from the eyes, flat-mounted onto glass slides, and postfixed for 60 minutes in 4% PFA at 4˚C.

## Proliferation assay in vivo

For in vivo analysis of cell proliferation, 5 μg of 5-ethynyl-2-deoxyuridine (EdU, C10337; Invitrogen) per gram of body weight was injected intraperitoneally 2 hours before sacrifice. Retinas were dissected and immunostained as described above. After secondary antibody staining, EdU labeling was detected by means of a Click-it EdU Alexa Fluor-488 Imaging Kit (C10337; Invitrogen) according to the manufacturer's instructions.

## Image acquisition and quantification

All fluorescence images were obtained using a confocal laser-scanning microscope (Leica TCS SP8, Leica Microsystems GmbH, Wetzlar, Germany). The image-based quantification methods are described in previous reports [28, 33–35]. The ratio of the avascular area to the whole retinal area (% avascular area), the ratio of the neovascular tufts area to the whole retinal area (% NVT area), the ratio of the Ter119$^+$ area to the whole retinal area (% Ter119$^+$ area), the distance from the optic disc to the vascular front (radial growth), and total pupillary vessel length were manually measured using ImageJ software by an investigator masked to treatment assignment. The sinuosity index of retinal vessels was calculated using the following formula: The sinuosity index = total curvilinear length of retinal vessels / total Euclidean distance between vascular branching points. The vessel density was calculated by using a binarization method. The col4$^+$CD31$^-$ area was considered as the area of regressed capillaries according to previous reports [36, 37], and the ratio of the col4$^+$CD31$^-$ area to the total col4$^+$ area (% area of regressed capillaries) was calculated. To quantify the number of endothelial cells (ECs), the EC-specific nuclear marker (Erg) was used and EdU$^+$Erg$^+$ cells were considered to be proliferating ECs [38].

## Western blot analysis

The constituent proteins in 100, 50, or 25 ng of Spodoptera frugiperda insect cell (Sf 21)-derived recombinant human VEGF 165 protein (R&D Systems, Inc., Minneapolis, MN, USA), Sf 21-derived recombinant mouse VEGF 164 protein (R&D Systems, Inc), E. coli-derived human VEGF 121 protein (R&D Systems, Inc), or E. coli-derived mouse VEGF 120 protein (R&D Systems, Inc) were separated by SDS-PAGE and transferred to polyvinylidene difluoride membranes (Bio-Rad Laboratories, Inc., Hercules, CA, USA). After blocking by 5% skim milk, the blots were incubated with aflibercept or ranibizumab at a final concentration of 1 μg/mL followed by anti-human IgG polyclonal antibody conjugated with horseradish peroxidase (for aflibercept) or an anti-human IgG F(ab')2 fragment-specific polyclonal antibody conjugated with horseradish peroxidase (for ranibizumab). Horseradish peroxidase activity was visualized using an enhanced chemiluminescence (ECL) detection system (FUJIFILM Wako Pure Chemical Corporation, Osaka, Japan).

## Statistical analysis

All statistical analyses were conducted using SPSS statistics (version 22; International Business Machines Co. Limited, New York, NY, USA). One-way analysis of variance was used for comparison between independent groups. A value of $P<0.05$ was considered statistically significant.

## Results

### No effect of intravitreal ranibizumab injection in OIR mice

To evaluate the neutralization of mouse VEGF by aflibercept and ranibizumab in vivo, we first examined retinal vascular structural changes after intravitreal injection of aflibercept and ranibizumab in oxygen-induced-retinopathy (OIR) model mice (Fig 1, n = 7 /group). When compared to PBS-injected eyes, aflibercept-injected eyes showed neovascular tufts (NVT) area reduction ($P = 0.02$, Fig 1C), avascular area enlargement ($P<0.01$, Fig 1D), and decreased sinuosity of retinal vessels ($P<0.01$, Fig 1E). In ranibizumab-injected eyes, however, the NVT area ($P = 1.00$, Fig 1C), avascular area ($P = 0.95$, Fig 1D), and sinuosity of retinal vessels ($P = 0.92$, Fig 1E) were comparable with those of PBS injected eyes.

Next, we observed retinas using fluorescein isothiocyanate (FITC)-dextran perfusion in OIR mice and found that retinal vascular permeability was suppressed in aflibercept-treated eyes, but not in ranibizumab-treated eyes (Fig 1F).

### No effect of multiple intraperitoneal ranibizumab injections in neonatal mice

As described above, no retinal vascular change was observed after a single intravitreal ranibizumab injection in the analysis using OIR model mice. However, some neutralization of mouse VEGF might be observed by increasing the dose or frequency of ranibizumab administration. Since it is difficult to administer a large amount of drug or multiple doses of drug via intravitreal injection, we conducted the examination using multiple intraperitoneal injection of a large amount of anti-VEGF drugs in neonatal mice (Fig 2A). Based on a previous report showing that intraperitoneal administration of 10 mg/kg of aflibercept in developing mice causes severe developmental defects [26], we injected 10 mg/kg of aflibercept intraperitoneally in the aflibercept group. Because 0.5mg of ranibizumab is usually administered for every 2 mg of aflibercept in human clinical practice, the low-dose ranibizumab group received 2.5 mg/kg of ranibizumab, corresponding to one-quarter of the dose of aflibercept. The high-dose ranibizumab group received 10 mg/kg of ranibizumab, corresponding to the full dose of aflibercept.

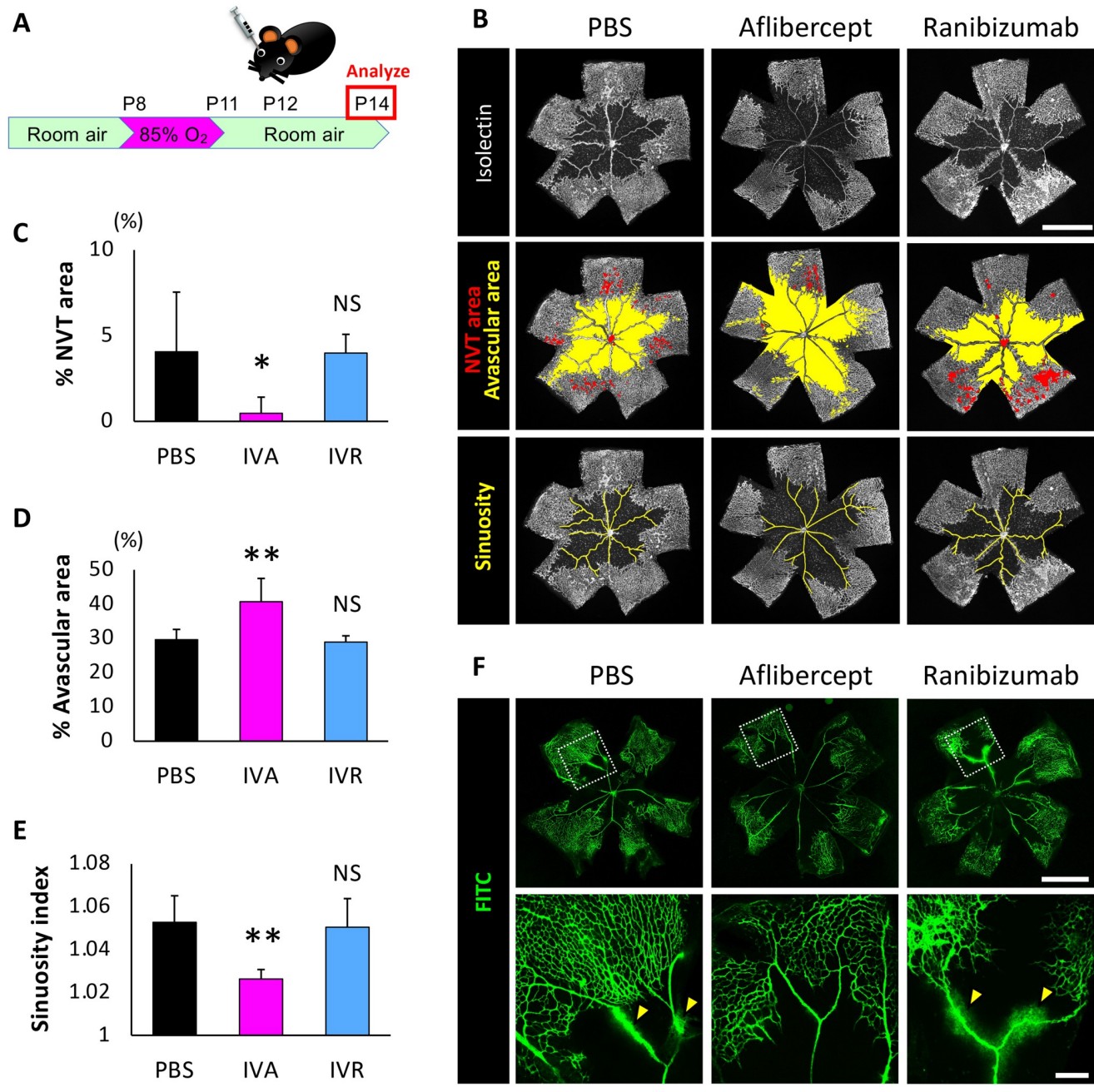

**Fig 1. The effect of intravitreal injection of ranibizumab and aflibercept on oxygen induced retinopathy.** (A) A schematic diagram depicting the time course of our experiment. (B) Whole-mounted stained retinas on day 2 after injection. (C) The reduction of the NVT area was observed in aflibercept injected eyes, but not in ranibizumab injected eyes. (D) Avascular area enlargement was observed in aflibercept injected eyes, but not in ranibizumab injected eyes. (E) Decreased sinuosity of the retinal vessel was observed in aflibercept injected eyes, but not in ranibizumab injected eyes. (F) Retinal vascular permeability was suppressed in aflibercept-treated eyes, but not in ranibizumab-treated eyes (arrowheads). Scale bars represent 1000 μm (B and upper row of F); 200 μm (lower row of F); $**P<0.01$; $*P<0.05$ (n = 7 /group). Data are presented as mean ± SD. PBS, phosphate-buffered saline solution; IVA, intravitreal aflibercept; IVR, intravitreal ranibizumab; NVT, neovascular tuft; NS, not significant.

After 4 intraperitoneal injections of PBS (n = 7), aflibercept (10 mg/kg, n = 6), or a low (2.5 mg/kg, n = 6) or high (10 mg/kg, n = 6) dose of ranibizumab, body weight gain in pups treated with aflibercept was significantly impaired comparing with other three groups ($P<0.01$, Fig 2B). The body weight gain in pups treated with 2.5 mg/kg or 10 mg/kg of ranibizumab was

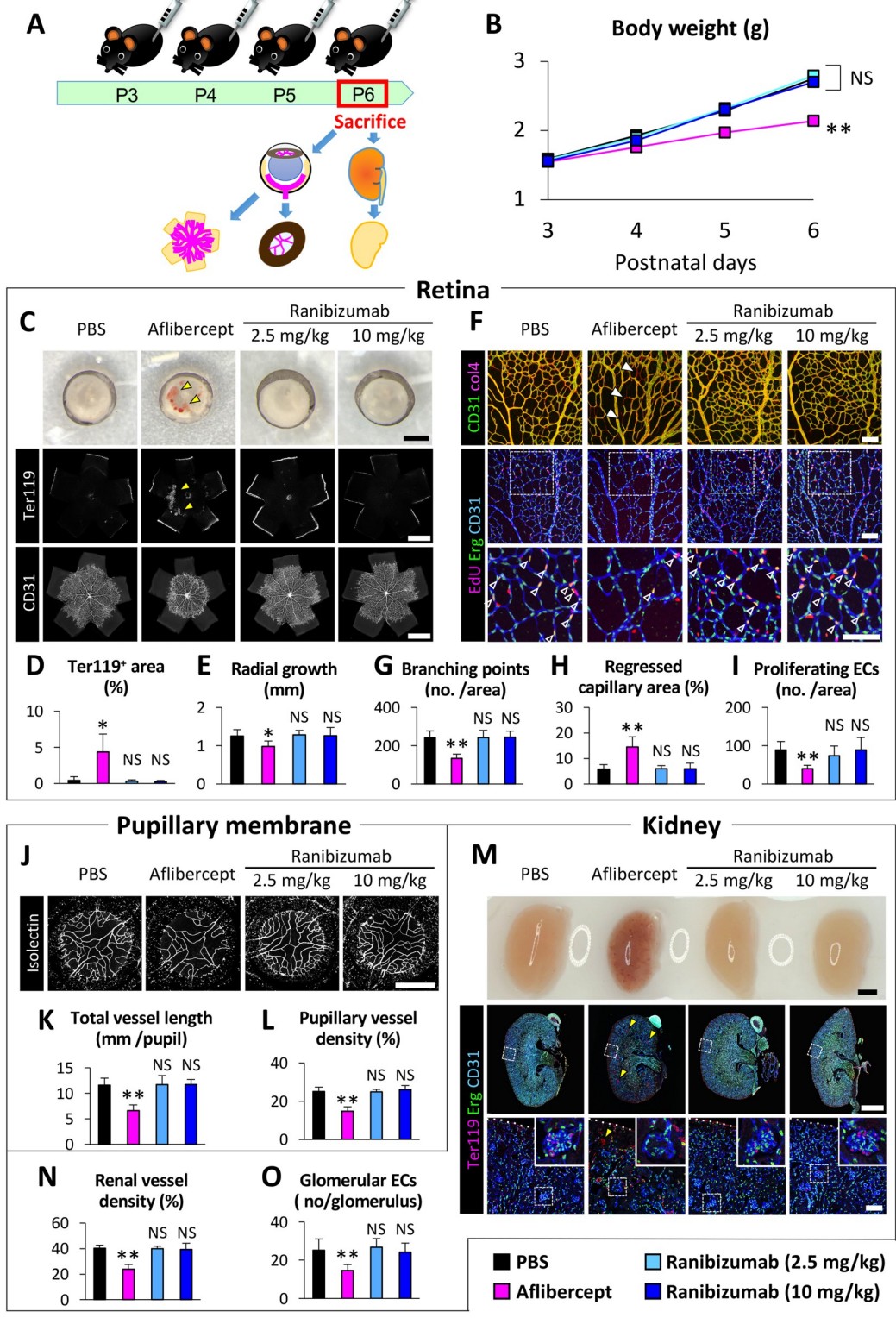

**Fig 2. The effect of ranibizumab and aflibercept injected intraperitoneally into neonatal mice.** (A) A schematic diagram depicting the time course of our experiment. (B) The body weight gains were impaired in the aflibercept group but not in the ranibizumab groups. (C-E) The whole-mounted retinas showed apparent retinal hemorrhage (arrowheads) and vascular growth impairment in the aflibercept group but not in the ranibizumab groups. (F-I) The high magnification images of the whole-mounted retinas show decreased number of branching points, increased area of regressed capillaries

(filled arrowheads), and decreased number of proliferating ECs (open arrowheads) in the aflibercept group but not in the ranibizumab groups. (J-L) The whole-mounted pupillary membranes showed decreased pupillary vessel length and density in the aflibercept group but not in the ranibizumab groups. (M-O) The kidney sections showed pathological renal thrombosis (arrowheads), decreased density of renal vessels, and decreased number of glomerular endothelial cells in the aflibercept group but not in the ranibizumab groups. Scale bars represent 1000 μm (C, top and middle rows of M); 500 μm (J); 100 μm (F and bottom row of M); $^{**}P<0.01$; $^{*}P<0.05$ (n = 6–7 /group). ECs, endothelial cells; NS, not significant; PBS, phosphate-buffered saline solution.

comparable with that in PBS treated pups ($P$ = 0.99). The whole-mounted retinas in the aflibercept group also showed apparent retinal hemorrhage (arrowheads in Fig 2C), increased Ter119$^+$ area ($P$ = 0.04, Fig 2D), retinal vascular growth impairment ($P$ = 0.04, Fig 2C and 2E), decreased number of branching points ($P<0.01$, Fig 2F and 2G), increased area of col4$^+$CD31$^-$ regressed capillaries ($P<0.01$, filled arrowheads in Fig 2F and 2H), and decreased number of EdU$^+$Erg$^+$ proliferating ECs ($P<0.01$, open arrowheads in Fig 2F and 2I), in contrast to the ranibizumab groups. Furthermore, decreased pupillary vessel length ($P<0.01$, Fig 2J and 2K) and density ($P<0.01$, Fig 2J and 2L), pathological renal thrombosis (arrowheads in Fig 2M), decreased renal vessel density ($P<0.01$, Fig 2M and 2N), and decreased number of glomerular ECs ($P<0.01$, Fig 2M and 2O) were observed in the aflibercept group, but not in the ranibizumab groups.

## No binding of ranibizumab to mouse VEGF-A in vitro

In vivo experiments showed that ranibizumab had no effect on mice when administered either by the intravitreal or systemic routes. To verify it in vitro, we performed Western blot analysis of the binding of human and mouse VEGF-A with aflibercept or ranibizumab as the primary antibody (Fig 3, S1 and S2 Figs). In the aflibercept-probed blot, immunoreactive bands of the expected size for both human and mouse VEGF-A were observed. On the other hand, in the ranibizumab-probed blot, immunoreactive bands were observed for human VEGF-A, but not for mouse VEGF-A even with long exposure time (10 min).

## Discussion

Both in vivo and in vitro, we did not observe any interaction between ranibizumab and mouse VEGF-A. This is a reasonable result because the structure of ranibizumab is similar to that of bevacizumab Fab, which lacks the ability to interact with mouse VEGF [29, 30]. Since there are many articles about the use of ranibizumab as an effective anti-VEGF drug in mice [17–23], we believe it is necessary to alert ophthalmology researchers against using ranibizumab as an anti-VEGF drug in the future mouse experiments.

It is unclear why ranibizumab showed anti-VEGF effects in previous reports [17–23], but we suspect that the use of poor quantitative methods such as fluorescein angiography [19], immunostaining of retinal sections [18–20], or FITC-dextran perfused retinal flat mounts [18, 20] may have led to different results. Fluorescein angiography is suitable for qualitative analysis, but not quantitative analysis because the amount of flash light and the timing of the photography after fluorescein injection may cause the results to vary. Immunostaining of retinal sections is also not suitable for quantitative analysis, especially in retinal vessels, because the quantitative results greatly depend on the slice position. FITC-dextran perfused retinal flat mounts have the following weakness: if the perfusion of FITC-dextran injected into the heart is inadequate, neovascular area may be underestimated and avascular area overestimated, so it is unsuitable for quantitative evaluation of vascular structure. Unlike these methods, immunostaining of whole-mounted retina is suitable for quantitative analysis because the

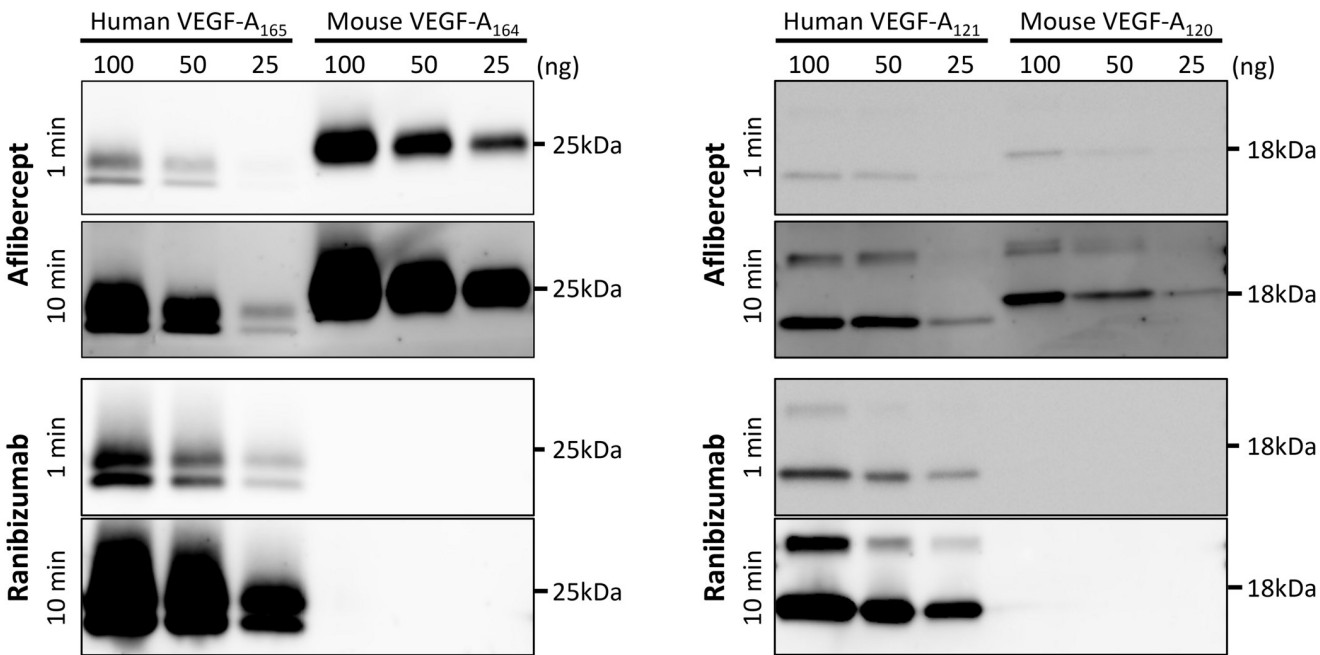

**Fig 3. Western blot analysis of the interaction of recombinant human VEGF-A$_{165}$, and mouse VEGF-A$_{164}$, human VEGF-A$_{121}$, and mouse VEGF-A$_{120}$ with aflibercept or ranibizumab as the primary antibody.** The immunoreactive bands for mouse VEGF-A were observed in the aflibercept-probed blot, but not in the ranibizumab-probed blot even with long exposure time. VEGF, vascular endothelial growth factor.

entire retinal vascular structure can be observed clearly [28, 34, 38]. In this study, we analyzed high quality images of whole-mounted retina and found ranibizumab had no effect on retinal vessels. Another possible reason for the difference between the present results and the results in the previous reports using ranibizumab may be the different amount of drug injected into the vitreous. We injected 5 μg of ranibizumab into the vitreous, but previous reports [18–21, 23] have used 1 μg to 10 μg. We cannot completely deny the possibility that ranibizumab might have had some effect on the OIR retina if we had used 10 μg in this study. However, considering that the volume of the mouse eye is about 1/400 of that of the human eye [39], 5 μg (0.5μL of Lucentis$^{\text{®}}$) was a sufficient dose, so we still believe that the drug effect of ranibizumab in mice is very small. In fact, 0.5 μL of Eylea$^{\text{®}}$ has a robust effect on the OIR retina, and it is clear that ranibizumab has much less effect on mice than aflibercept.

In the in vivo experiments of this study, we first analyzed change in retinal vascular structure after intravitreal injection of anti-VEGF drugs using OIR mice and found no anti-VEGF effect of ranibizumab in OIR retina. In this method using OIR model mice, however, small anti-VEGF effects may be missed because of a large variation in the degree of retinopathy between individual OIR model mice and because of the dosage limit in ocular administration. In order to capture even a small anti-VEGF effect, we conducted an additional experiment in which an extremely large dose of anti-VEGF drug was administered intraperitoneally to neonatal mice because there is little variation in vascular development between individual neonatal mice. In this additional experiment, we administered 10 mg/kg (87 nmol/kg) of aflibercept, or 2.5 mg/kg (52 nmol/kg) or 10 mg/kg (208 nmol/kg) of ranibizumab for 4 consecutive days, which are much higher doses than the dose of aflibercept used as an anti-cancer drug in humans (35 nmol/kg every two weeks). However, multiple administration of extremely large doses of ranibizumab to neonatal mice did not cause any change compared to PBS administration. Although our results could be influenced by the difference in the clearance rate from the

circulation between ranibizumab and aflibercept [40], we believe our results still suggest that ranibizumab does not neutralize mouse VEGF.

A limitation of this study is that we only analyze the OIR retina at 2 days after intravitreal injection of ranibizumab. We used this method because in our previous study [28], we analyzed OIR retinas 2 days and 4 days after intravitreal aflibercept injections and found that the anti-VEGF effects of aflibercept were stronger on P14 than on P16. However, it is unclear whether ranibizumab and aflibercept have the same pharmacokinetics in the mouse eye, and we cannot rule out the possibility that a slight anti-VEGF effect of ranibizumab could have been detected if the analysis had been performed at another time point after ranibizumab injection. The other limitation is lack of sophisticated binding assays which can quantitatively assess the binding affinity of aflibercept and ranibizumab to mouse VEGF. However, the main purpose of this study was to investigate the neutralizing efficacy of ranibizumab and aflibercept on mouse VEGF in vivo, and we showed that ranibizumab lacks the ability to neutralize mouse VEGF mainly in vivo. Although this study did not quantitatively assess the binding affinity of aflibercept and ranibizumab to mouse VEGF, we believe our data that ranibizumab is not suitable for in vivo experiments in mice provides ophthalmology researchers with important information.

In conclusion, ranibizumab does not neutralize mouse VEGF, both in vivo and in vitro. When conducting experiments using anti-VEGF drugs in mice, aflibercept is suitable, but ranibizumab is not.

## Supporting information

**S1 Fig. Full-length western blot images of recombinant human VEGF-A$_{165}$ and mouse VEGF-A$_{164}$ detected with aflibercept or ranibizumab as the primary antibody.** Immunoreactive bands of mouse VEGF-A$_{164}$ were observed in the aflibercept-probed blot, but not in the ranibizumab-probed blot. VEGF, vascular endothelial growth factor.
(TIF)

**S2 Fig. Full-length western blot images of recombinant human VEGF-A$_{121}$ and mouse VEGF-A$_{120}$ detected with aflibercept or ranibizumab as the primary antibody.** Immunoreactive bands of mouse VEGF-A$_{120}$ were observed in the aflibercept-probed blot, but not in the ranibizumab-probed blot. VEGF, vascular endothelial growth factor.
(TIF)

**S1 Raw images.**
(PDF)

## Acknowledgments

We thank Fumiko Kimura for her technical support.

## Author Contributions

**Conceptualization:** Yusuke Ichiyama, Masahito Ohji.

**Data curation:** Yusuke Ichiyama, Riko Matsumoto.

**Formal analysis:** Yusuke Ichiyama, Shumpei Obata, Osamu Sawada, Yoshitsugu Saishin, Masashi Kakinoki, Tomoko Sawada.

**Funding acquisition:** Yusuke Ichiyama.

**Investigation:** Yusuke Ichiyama, Riko Matsumoto, Shumpei Obata, Masashi Kakinoki.

**Methodology:** Yusuke Ichiyama, Riko Matsumoto, Osamu Sawada, Yoshitsugu Saishin.

**Project administration:** Yusuke Ichiyama.

**Supervision:** Masahito Ohji.

**Writing – original draft:** Yusuke Ichiyama.

**Writing – review & editing:** Masahito Ohji.

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
