## [Decision Letter · Decision Letter 0]

17 Aug 2022

PONE-D-22-19258Assessment of mouse VEGF neutralization by ranibizumab and afliberceptPLOS ONE

Dear Dr. Ichiyama,

Thank you for submitting your manuscript to PLOS ONE. After careful consideration, we feel that it has merit but does not fully meet PLOS ONE’s publication criteria as it currently stands. Therefore, we invite you to submit a revised version of the manuscript that addresses the points raised during the review process.

both reviewers found this an important article and made some minor comments that can be easily addressed . we look forward to the revised manuscript

We look forward to receiving your revised manuscript.

Kind regards,

Demetrios G. Vavvas

Academic Editor

PLOS ONE

Journal Requirements:

2. Please clarify in the Methods section and Ethics statement all methods of anesthesia and euthanasia employed in the study.

"We thank Fumiko Kimura for her technical support.

This work was supported by grants-in-aid for Scientific Research on Innovative Areas from the Japan Society for the Promotion of Science (19K18877). This funding organization had no role in the design or conduct of this study".

 "This work was supported by grants-in-aid for Scientific Research on Innovative Areas from the Ministry of Education, Culture, Sports, Science, and Technology of Japan (19K18877). This funding organization had no role in the design or conduct of this study".

Reviewers' comments:

Reviewer's Responses to Questions

**Comments to the Author**

1. Is the manuscript technically sound, and do the data support the conclusions?

Reviewer #1: Yes

Reviewer #2: Yes

2. Has the statistical analysis been performed appropriately and rigorously? 

Reviewer #1: Yes

Reviewer #2: Yes

3. Have the authors made all data underlying the findings in their manuscript fully available?

Reviewer #1: Yes

Reviewer #2: Yes

4. Is the manuscript presented in an intelligible fashion and written in standard English?

Reviewer #1: Yes

Reviewer #2: Yes

5. Review Comments to the Author

Reviewer #1: The authors use different models of in vitro and in vivo experiments to show that Aflibercept, and not Ranibizumab, interacts significantly with mouse VEGF. This could significantly impact translational research investigating neovascular proliferation and retinal vasculature development. However, some clarification is needed:

1) Authors state that the image-based quantification on this study was made manually with Image J. Please clarify if the investigator working on the images was blinded, since the manual processing could be biased. If any plugin or macro was used, please describe it on “Materials and Methods”. If the investigator was not blinded, this should be stated

2) Please clarify if PBS was injected on the fellow eye of anti-VEGF injected eyes. The control should be the fellow eye. If that was the case, please separate the PBS column on Figure 1-C, 1-D, 1-E into PBS-injected eyes from Aflibercept treated mice and Ranibizumab treated mice.

Reviewer #2: This is a very interesting study presenting the results of primary scientific work and reporting on mouse VEGF neutralization by ranibizumab and aflibercept both in vivo and in vitro. The authors managed to present a sound and thoroughly organized study and relate their findings well to previous studies. However, there are a few points that need to be addressed.

Major points

Methods: Details regarding how many mice were used in total and for each experiment should be provided.

In addition, the authors should provide references for the protocols they used for each scientific method and experiment they conducted. (e.g. 87-90, 105-109: The protocols used are missing.)

Lines 100-103: The previously described finding that anti-VEGF effects of aflibercept were stronger on P14 could not stand for ranibizumab, too, since it has not been investigated earlier. This fact could affect the findings of the present study and should also be included as a limitation in the discussion section (lines 269-270).

Lines 105-109, 202-205: The authors should also add references to support the selection of this procedure, otherwise provide a more detailed explanation.

Lines 121, 157: Furthermore, they had better clarify which anti-collagen antibody, i.e. coll V or coll IV, they used.

The authors should include captions for the providing figures and details regarding what each part of the figure, e.g. 1A, 1B, 1C, depicts.

I would like to look at a revised version of the manuscript.

6. PLOS authors have the option to publish the peer review history of their article (what does this mean?). If published, this will include your full peer review and any attached files.

Reviewer #1: No

Reviewer #2: No

---

## [Author Response · Author response to Decision Letter 0]

30 Sep 2022

Response to Reviewer #1:

Authors state that the image-based quantification on this study was made manually with Image J. Please clarify if the investigator working on the images was blinded, since the manual processing could be biased. If any plugin or macro was used, please describe it on “Materials and Methods”. If the investigator was not blinded, this should be stated.

Response: Thank you for your comment. The quantification was performed by a masked investigator. We revised the "Materials and Methods" section as follows.

Page 11, line 153 (Result)

The ratio of the avascular area to the whole retinal area (% avascular area), the ratio of the neovascular tufts area to the whole retinal area (% NVT area), the ratio of the Ter119+ area to the whole retinal area (% Ter119+ area), the distance from the optic disc to the vascular front (radial growth), and total pupillary vessel length were manually measured using ImageJ software by an investigator masked to treatment assignment.

Please clarify if PBS was injected on the fellow eye of anti-VEGF injected eyes. The control should be the fellow eye. If that was the case, please separate the PBS column on Figure 1-C, 1-D, 1-E into PBS-injected eyes from Aflibercept treated mice and Ranibizumab treated mice.

Response: Thank you very much for pointing out this important point. We previously reported that intravitreally injected aflibercept migrates hematogenously to the fellow eye, where it exerts its anti-VEGF effect. In this study, therefore, we did not use the fellow eyes of the anti-VEGF injected eyes as control group. To minimize the effects of individual differences, littermates were divided into PBS, aflibercept, and ranibizumab groups. We added the following sentences to "Materials and Methods" section.

Page 8, line 103 (Result)

To minimize the effects of individual differences, littermates were divided into the above three groups.

Page 8, line 111 (Result)

To minimize the effects of individual differences, littermates were divided into the above four groups.

Response to Reviewer #2:

Methods: Details regarding how many mice were used in total and for each experiment should be provided. In addition, the authors should provide references for the protocols they used for each scientific method and experiment they conducted. (e.g. 87-90, 105-109: The protocols used are missing.)

Lines 105-109, 202-205: The authors should also add references to support the selection of this procedure, otherwise provide a more detailed explanation.

Response: Thank you for pointing out these omissions. The numbers of mice used in total and for each experiment were added to our manuscript. In addition, references to the experimental methods used are indicated, and a detailed description is added as follows.

Page 7, line 86 (Materials and Methods)

Euthanasia for tissue sample collection was achieved by performing cardiac puncture exsanguination under isoflurane anesthesia. The right atrium was then pierced and 5 mL of phosphate-buffered saline solution (PBS) was perfused through the left ventricle. Right after, 5 mL of 4% paraformaldehyde (PFA) at 4 °C was perfused in the same manner to start fixation of the tissues [28]. A total of 46 mice were used for this study.

Page 16, line 226 (Result)

Based on a previous report showing that intraperitoneal administration of 10 mg/kg of aflibercept in developing mice causes severe developmental defects [26], we injected 10 mg/kg of aflibercept intraperitoneally in the aflibercept group. Because 0.5 mg of ranibizumab is usually administered for every 2 mg of aflibercept administered in human clinical practice, the low-dose ranibizumab group received 2.5 mg/kg of ranibizumab, corresponding to one-quarter of the dose of aflibercept. The high-dose ranibizumab group received 10 mg/kg of ranibizumab, corresponding to the full dose of aflibercept.

Lines 100-103: The previously described finding that anti-VEGF effects of aflibercept were stronger on P14 could not stand for ranibizumab, too, since it has not been investigated earlier. This fact could affect the findings of the present study and should also be included as a limitation in the discussion section (lines 269-270).

Response: Thank you very much for your important point. We added following sentences in the “Discussion” section.

Page 22, line 333 (Discussion)

A limitation of this study is that we only analyze the OIR retina at 2 days after intravitreal injection of ranibizumab. We used this method because in our previous study [28], we analyzed OIR retinas 2 days and 4 days after intravitreal aflibercept injections and found that the anti-VEGF effects of aflibercept were stronger on P14 than on P16. However, it is unclear whether ranibizumab and aflibercept have the same pharmacokinetics in the mouse eye, and we cannot rule out the possibility that a slight anti-VEGF effect of ranibizumab could have been detected if the analysis had been performed at another time point after ranibizumab injection.

Lines 121, 157: Furthermore, they had better clarify which anti-collagen antibody, i.e. coll V or coll IV, they used.

Response: Thank you for your comment. We detected type IV collagen using an anti-collagen IV antibody. To avoid misunderstanding, we changed the abbreviation of type IV collagen from colIV to col4.

The authors should include captions for the providing figures and details regarding what each part of the figure, e.g. 1A, 1B, 1C, depicts.

Response: Thank you for your comment. We had listed figure captions at the end of the manuscript under the title "Legends". To meet PLOS ONE's style requirements, we relocated each figure caption to the position directly after the paragraph in which it was first cited. In addition, annotations were added to make it easier to understand what Fig 1B and 1F represent.

---

## [Decision Letter · Decision Letter 1]

25 Nov 2022

Assessment of mouse VEGF neutralization by ranibizumab and aflibercept

PONE-D-22-19258R1

Dear Dr. Ichiyama,

We’re pleased to inform you that your manuscript has been judged scientifically suitable for publication and will be formally accepted for publication once it meets all outstanding technical requirements.

Kind regards,

Demetrios G. Vavvas

Academic Editor

PLOS ONE

Additional Editor Comments (optional):

Reviewers' comments:

Reviewer's Responses to Questions

**Comments to the Author**

1. If the authors have adequately addressed your comments raised in a previous round of review and you feel that this manuscript is now acceptable for publication, you may indicate that here to bypass the “Comments to the Author” section, enter your conflict of interest statement in the “Confidential to Editor” section, and submit your "Accept" recommendation.

Reviewer #1: All comments have been addressed

Reviewer #2: All comments have been addressed

2. Is the manuscript technically sound, and do the data support the conclusions?

Reviewer #1: Yes

Reviewer #2: Yes

3. Has the statistical analysis been performed appropriately and rigorously? 

Reviewer #1: Yes

Reviewer #2: Yes

4. Have the authors made all data underlying the findings in their manuscript fully available?

Reviewer #1: Yes

Reviewer #2: Yes

5. Is the manuscript presented in an intelligible fashion and written in standard English?

Reviewer #1: Yes

Reviewer #2: Yes

6. Review Comments to the Author

Reviewer #1: The authors addressed all my comments on the original manuscript. Overall the quality of the manuscript has improved. I don't have any other concerns. I believe the manuscript is ready for publication.

Reviewer #2: (No Response)

7. PLOS authors have the option to publish the peer review history of their article (what does this mean?). If published, this will include your full peer review and any attached files.

Reviewer #1: No

Reviewer #2: No

---

## [Editor Report · Acceptance letter]

13 Dec 2022

PONE-D-22-19258R1 

Assessment of mouse VEGF neutralization by ranibizumab and aflibercept 

Dear Dr. Ichiyama:

I'm pleased to inform you that your manuscript has been deemed suitable for publication in PLOS ONE. Congratulations! Your manuscript is now with our production department. 

Kind regards, 

on behalf of

Prof. Demetrios G. Vavvas 

Academic Editor

PLOS ONE